# The Relationship between the Static and Dynamic Balance of the Body, the Influence of Eyesight and Muscle Tension in the Cervical Spine in CAA Patients—A Pilot Study

**DOI:** 10.3390/diagnostics11112036

**Published:** 2021-11-03

**Authors:** Anna Olczak, Aleksandra Truszczyńska-Baszak, Katarzyna Gniadek-Olejniczak

**Affiliations:** 1Rehabilitation Clinic, Military Institute of Medicine, 04-141 Warsaw, Poland; kgniadek-olejniczak@wim.mil.pl; 2Faculty od Medical Sciences, Social Academy of Science, 00-842 Warsaw, Poland; 3Faculty of Rehabilitation, Józef Piłsudski University of Physical Education in Warsaw, 00-968 Warsaw, Poland; aleksandra.truszczynska@awf.edu.pl

**Keywords:** stroke, CAA (Cerebral Amyloid Angiopathy), balance, muscle tension, vision

## Abstract

Cerebral amyloid angiopathy (CAA) is one form of disease of the small vessels of the brain and can cause frequent cerebral hemorrhages as well as other types of stroke. The aim of the study was to analyze the static and dynamic balance of the body and changes in the tension of selected muscles of the cervical spine in patients with CAA after stroke, depending on visual control or its absence, compared to healthy volunteers. Eight stroke patients and eight healthy subjects were examined. The functional Unterberger test and the Biodex SD platform were used to test the dynamic equilibrium, on which the static equilibrium was also assessed. Muscle tension was tested with the Luna EMG device. In static tests, the LC muscle (longus colli) was significantly more active with and without visual control (*p* = 0.016; *p* = 0.002), and in dynamic tests, significantly higher results for MOS (*p* = 0.046) were noted. The comparison of the groups led to the conclusion that the more functional deficits, the more difficult it is to keep balance, also with eye control.

## 1. Introduction

Cerebral amyloid angiopathy (CAA) is one of the forms of diseases of the small vessels of the brain. It causes the deposition of β-amyloid protein in the wall of arterioles, capillaries and venous vessels located in the cerebral cortex and the soft dura. The prevalence of CAA is very age dependent. From 2.3 percent in patients aged 65 to 74 years, from 12.1 percent in patients aged 85 and over [1]. Cerebral amyloid angiopathy is asymptomatic or with symptoms such as intracerebral hemorrhage, cognitive impairment or dementia [2,3]. Similar to Alzheimer’s disease, it is a degenerative disease of the group of diseases called amyloidopathy. It is a disease of old age and is the second most common (after atherosclerosis) cause of recurrent cerebral haemorrhages [4,5,6,7,8,9,10,11,12,13,14,15,16,17,18]. CAA is relatively most common in the occipital, frontal, temporal and parietal lobes. The cerebellum is most rarely involved, and it is practically absent in the basal ganglia, thalamus, white matter and brainstem. CAA is characterized by a variable course and a chronically progressive disease process [19,20].

In our work, we present a group of stroke patients diagnosed with CAA. Stroke as well as other neurological conditions such as Alzheimer’s disease, Parkinson’s disease and syndrome, peripheral neuropathies, epilepsy, migraine, multiple sclerosis, etc., are disorders related to imbalance. It should also be remembered that aging has a large effect on the equilibrium system [7,21,22].

Balance disorders are one of the most common problems in adults (affecting 20–30% of the population) [23,24,25]. Of this, women and people from the geriatric group report the problem twice as often [26,27].

Taking into account that the major functional problem of our patients were imbalances, we decided to check whether the organ of vision plays a significant role in stabilizing the body posture, both in static and in movement, and how it affects changes in the tension of selected muscles in the cervical spine. Turning off the eye control shows the effectiveness of the other balance control mechanisms. Researchers investigated the importance of muscle tone in the cervical spine. For example, the Koskimies et al. showed that balance can be altered in healthy people by stimulating the proprioceptors of the neck, hips or lower legs with vibration. More precisely, the study showed that in patients with neck tightness, postural stability was worse than in healthy subjects [28]. This was confirmed by studies showing that not only neck trauma, but also neck pain and muscle tension may influence the sensitivity of neck proprioception [29]. Moreover, Pyykkö et al. showed that there was a maximum response to vibration-induced changes in balance in the neck muscles compared to other muscles of the spine or lower limbs [30]. Small internal, deep dorsal and suboccipital muscles show a high density of muscle spindles [31], which are involved in neck proprioception [32].

According to Djupsjöback et al., cervical muscles deep fatigue may interfere with posture control, possibly due to the stimulation of tonic gamma motor neurons, secondary to the accumulation of muscle contraction metabolites [33]. These contractile metabolites, including potassium ions, lactic acid and arachidonic acid, stimulate afferent groups III and IV, which have the potential to initiate a positive feedback loop for increased muscle spindle sensitivity and thus activity via the gamma motor system [34]. The use of surface electromyography (EMG) to study functional abilities is well known. The frequency median (MF) appears to be a useful objective measure for assessing muscle function and fatigue in the cervical muscles. During the study of static and dynamic equilibrium [35], which we used in our research. In addition, we used the tandem test, the Unterberger test, and the static and dynamic balance assessment on the Biodex SD balance platform.

In practice, the tandem test is considered feasible and useful in neurological examination to assess the imbalance [36,37,38].

The Unterberger test of walking in place was proposed by Hickey et al. as a useful indicator of peripheral labyrinth dysfunction. However, the group of subjects suspected of decompensated peripheral labyrinth dysfunction was disproportionately smaller than the compared group of healthy volunteers. Perhaps that is why the study did not show a significant difference in the performance of the Unterberger step test between patients with disorders and healthy patients [39].

Greater reliability of the Unterberger (UT) test results, according to some researchers [40], may take place when the test is performed in conditions where there is a chance to remove various external stimuli (clocks, fans and telephones) and [41] perform the test in silence. Researchers claim that the localization of sound and surroundings can act as a spatial locator and influence the scope of rotation [38]. Moreover, scientists agree that UT alone should not be used as a screening instrument [36,42], but as an additional procedure to investigate imbalance as a symptom. As a result, the test can increase the amount of information that needs to be included in conjunction with the diagnostic test. Therefore, we included this simple test in our study.

We assessed the influence of the eye organ on the ability to maintain balance and changes in muscle tone of the cervical spine in patients with CAA and healthy volunteers, both with eyes open and closed. Similarly, Ana Cecília Grilli Fernandes et al. Studied healthy adults and elderly people with dizziness and/or vertigo, but with no clear evidence of impairment in the central nervous system, with the eyes open and closed with the Unterberger Fukuda test (UFT) [42].

A large number of scientific reports concern etiopathogenesis, diagnosis and clinical examination of cerebral amyloid angiopathy, but not the functional examination of such patients [43,44,45].

Literature search did not reveal any similar studies assessing body balance in CAA stroke patients, therefore our study focused on the study of body balance and the importance of the visual organ, as well as changes in cervical spine muscle tone in CAA patients compared to healthy controls.

The aim of the study was to assess the ability to maintain static and dynamic balance of the body and the distribution of muscle tension in the cervical spine, depending on the eye control, in CAA patients and healthy volunteers.

## 2. Material and Methods

### 2.1. Study Design

This is an observational study. The static equilibrium was assessed on a Biodex SD balance platform (stability test). The dynamic equilibrium was tested in the Unterberger test and on the Biodex SD balance platform (the fall risk test). All examinations were performed twice with eyes open and then closed. It was investigated how the intervention, control and the lack of eye control (independent variables), influence the displacement of the center of body mass in the conditions of statics and dynamics, and how changes the tension of selected muscles in the cervical spine (dependent variables).

A group of healthy people matched for gender, age, weight and height was tested to assess whether neurological deficits in post-stroke patients could affect static and dynamic balance results and changes in muscle tone, and whether the visual organ plays a large role in controlling the center of mass body and changes in muscle tension, both sick and healthy.

### 2.2. Ethics

The study was carried out in the Teaching Department of Rehabilitation of the Military Medical Institute (MMI) in Warsaw, Poland. It was approved by and carried out in accordance with the recommendations of the Ethical Committee of the Military Medical Institute (MMI; approval number 5/MMI/2020). Prior to inclusion, all subjects were informed about the purpose of the study. Written informed consent was obtained from all subjects in accordance with the tenets of the Declaration of Helsinki.

### 2.3. Patients and Subjects

The study was conducted at the Military Medical Institute in Warsaw in the Rehabilitation Clinic with the Neurological Rehabilitation Sub-Unit and the Day Rehabilitation Department from October 2019 till September 2020. The study was registered in the international Clinical Trial database under number NCT 05082194.

The study involved 16 people. The National Institute for Health Stroke Scale (NIHSS) [46] was used to identify the neurological deficit, and the modified Rankin scale [47,48,49] was used to evaluate the patients’ overall physical impairment. In addition, the Tandem Test (Tandem Stance and Walk Test) was used to assess the imbalance in standing and walking.

Main criteria for stroke group inclusion: (1) purposeful selection of patients after CAA stroke; (2) the functional state of the patients, which allows independent walking in a place and at a distance (Rankin Scale ≤ 3); (3) muscle tension Modified Ashworth Scale (MAS ≤ 1/1 +); (4) no severe deficits in communication, memory, or understanding what can impede proper measurement performance; (5) at least 55 years of age; maximum 90 years old.

Main criteria for stroke group exclusion: (1) cause of stroke other than CAA; (2) stroke up to seven weeks after the episode; (3) epilepsy; (4) lack of trunk stability; (5) lack of independent walking; (6) high or very low blood pressure, dizziness, malaise.

Main criteria for control group inclusion: (1) purposeful selection of healthy people for patients, taking into account sex, age, body weight and height; (2) the control group consisted of healthy subjects with stable trunk (TCT 100 points); correct muscle tension (MAS = 0), independent walking; (3) at least 55 years of age; maximum 90 years old.

Criteria for control group exclusion: (1) a history of neurologic or musculoskeletal disorders such as stroke or brain injury or other conditions that could affect their ability to active movement the trunk and the legs; (2) pain, dizziness; (3) permanent use of orthopedic supplies; (4) severe deficits in communication, memory, or understanding what can impede proper measurement performance; (5) high or very low blood pressure, dizziness, malaise.

Patients after a CAA stroke were selected from a group of 30 other stroke patients who met the criteria of static and dynamic balance and the tension of selected muscles of the cervical spine with and without visual control. Similarly, a group of healthy people was selected from the group of 30 healthy people, which in the above project was the control group. Healthy people were purposefully selected and matched in terms of sex, age, height and weight.

Ultimately, 8 post-stroke patients diagnosed with CAA (mean age 65.38 ± 11.34) and 8 healthy people (mean age 64.50 ± 9.83) participated in the study. Study group was past stroke 7–9 weeks, with stable trunk (the Trunk Control Test 70–100 points) [50,51], muscle tension measured with Modified Ashworth Scale (MAS = 1/1 +) [52], unassisted walking (modified Rankin scale = 3), with slight neurological deficits (NIHSS ≤ 7). In this group, the Tandem Test, both standing and walking, was positive. The control group was highly functional, Rankin scale = 0, without pain, with stable trunk (the Trunk Control Test 100 points), tension of muscles measured with Modified Ashworth Scale (MAS = 0) [50,51,52]. In this group, the Tandem Test, both standing and walking, was negative. The clinical evaluation of patients after a stroke was performed by the physician admitting the patient to the clinic on the day of admission. The epidemiological data of post stroke population and healthy control groups is in Table 1.

At the same time, there were no significant differences between the groups in terms of age, sex, weight, height, or BMI (Table 2).

### 2.4. Functional Assessment Tools

In order to assess the functional balance of patients and the influence of eyesight on the balance and muscle tone of the cervical spine, the following tools were used in the study: Tandem Stance Test, Tandem Walk Test, Modified Unterberger Attempt, Posture stability research on the Biodex SD platform, Fall risk test on the Biodex SD platform, assessment of muscle tone by superficial EMG using the Luna EMG device (EGZOTech, Gliwice, Poland).

Tandem tests are qualitative tests, in our research we used them to initially assess the stability of the posture in the conditions of statics and dynamics.

The Tandem Test consists of 3 tasks, but for the purposes of our research, we used two of them: Tandem Stance Test and Tandem Walk Test. The Tandem Stance Test consists in taking the patient in an upright standing position, slightly stretching so that the feet are at hip level, lifts the right foot over the left one and keeps this position for a few seconds, returns to the starting position, and then repeats the same action with left foot. The Tandem Walk Test involves putting yourself in an upright standing position and walking in a straight line foot by foot. For ease of use, a line or tape is drawn on the floor.

The Unterberger functional test is also included in the qualitative tests. The Unterberger Attempt is to walk with your eyes closed and your upper limbs stretched out in front of you. Execution: starting position of the examined person, standing. The examiner’s starting position: standing at a distance so as not to disturb the performance of the test. Procedure: the subject with eyes closed performs 50 steps with high knee lift. Interpretation: under normal conditions, the test person only slightly changes the direction of walking in place (a deviation from the original direction by 45° is allowed). A clear change of direction speaks in favor of the labyrinthine origin of imbalances. Unilateral or bilateral stagger points to the cerebellar basis of the imbalance. For the purposes of the study, the first attempt was made with eyes open and the next with eyes closed.

The Biodex Balance System™ SD (Biodex Medical Systems, Inc., Shirley, NY, USA) balance platform was used to test the postural stability and the risk of falling. This platform enables testing and training in both static and dynamic formats. It is a system that provides a quick and accurate fall risk assessment and tests your balance in the static. Biofeedback—enables real-time feedback, which is an important element of the neuromuscular re-education process. Additionally, it enables the use of a protocol covering the risk of falling, the limits of stability and the posture of stability [53,54].

The degree of surface instability is controlled by the microprocessor-based actuator of the SD system (from 0 to 12 levels = up to 20 degrees of deflection in each direction, counting from the level). The clinician selects the test duration, stability level, and protocol. In the dynamic test, when the test begins, the tilt angle of the platform is quantified as the deviation from the locked (horizontal) position as well as the degrees of the yaw over time. Further insight into specific neuromuscular activation patterns is gained by quantifying the anterior/posterior and mid/lateral inclination of the platform. After testing, predictive values and comparative (two-sided) reports are available to plot the patient’s stability/stability/fall risk performance. Two-sided comparisons document the differences between each leg. Static tests measure the angular deviation of the patient’s center of gravity. In the case of static measurements, body height should be taken into account. Based on the selected height, the appropriate scaling of the static measures is applied. Testing in this mode is ideal for basic testing for movement disorders, vestibular dysfunction, and orthopedic patients. Good results in static tests may lead to the transition to dynamic testing and training [53,54].

The Luna EMG rehabilitation device was used to assess muscle tone.

A Luna EMG (a rehabilitation-diagnostic robot developed by EGZOTech, Gliwice, Poland) was used to measure muscle tension (accuracy of measurement [−1 ± 1 V ± 1 µV]). Surface electrodes (single-use 55 × 40 mm; ECG Electrodes; Sorimex, Poland) were glued to the subject’s body according to the SENIAM (Surface ElectroMyoGraphy for the Non-Invasive Assessment of Muscles) procedure for the sternocleidomastoid and longus colli.

### 2.5. Measurements

The research was carried out according to protocol no 8/KRN/2020, registered in Clinical Trial Registration. Each of the examined persons performed the given test twice, with the eyes open the first time and the eyes closed the second time. Moreover, both during the Unterberger functional test and the tests on the Biodex SD balance platform, the muscle tone was assessed in the subjects using the Luna EMG device.

Before starting the tests, surface electrodes were glued to the patient’s body (according to the SENIAM procedure). Five surface electrodes were used, consisting of three channels. Two first channel electrodes were placed on the abdomen of the right sternocleidomastoid muscle, two more were placed on the left longus colli muscle and one reference electrode on the right shoulder process of the scapula. The sternocleidomastoid muscle is superficial, and the palpation location did not cause any problems. The patient’s task was to move the head to the right, twist it to the opposite side and lift the face upwards. In this position, this muscle was visible, which allowed for the sticking of two electrodes on his abdomen. In turn, the long muscle of the neck is a deep muscle and is involved in the flexion of the cervical spine. The patient performed head flexion, which made it possible to stick another two electrodes on the left side of the cervical vertebrae according to the SENIAM procedure. After each completed test, an ongoing test report was generated, which showed the results for both muscles (voltage values in microvolts [µV]). For statistical analysis, the mean value of the tension of the examined muscles was taken.

The Unterberger test for the purposes of this study was modified and divided into two trials. The subjects performed the test once with their eyes open and then with their eyes closed. The participants’ task was to assume a standing position with legs apart at hip width, and then walk in place (50 steps). Moreover, the modification of the test consisted in the fact that the patients during these tests performed the test with the upper limbs along the body as when walking.

Static and dynamic balance testing on the Biodex SD balance platform.

Testing stability of attitude with the use of the Biodex SD balance platform. In order to use the platform, the patient must first be positioned on it and the parameters concerning: placement of the feet, age and height of the examined person are saved on the platform. The platform remains stable throughout the test. Test duration (20 s), number of attempts (3), with ten-second rest periods between each attempt. The examined person stands both with feet shoulder-width apart; the medial ankles of the subject protruded 5 cm in relation to the geometric center of the platform, marked on its surface [53,54]. For statistical analysis, the bariums were the mean results of the three trials (calculated by the device) for both the eyes open and eyes closed testing. The patient’s score in this test evaluates deviations from the center, so a lower score is more desirable than a higher score.

Finally, the dynamic equilibrium in the Fall Risk test was also assessed on the Biodex SD platform. During the study, the patient underwent three trials of 20 seconds each, starting with the initial platform set at level 6 and ending with the platform set at level 2, with ten-second rest periods between each trial. The patient’s setting should not change. Thus, the position of the examined person was used, checked and, if necessary, corrected in accordance with the parameters of the stop setting from the posture stability test [53,54]. Similar to the statistical analysis, the barns were the mean results of the three trials (calculated by the device), both for the testing with eyes open and closed.

In both tests, the subject performed each of the tests 6 times, 3 with eyes open and 3 with eyes closed.

### 2.6. Sample Size Calculation

The test power was estimated based on the available data. Sample size—for repeated measurements (comparison of samples), assuming that we have a power of 0.8; alpha = 0.05 and the effect size of 1.03 is the minimum sample size required to be 8. The effect size was estimated from means, standard deviation, and correlation between measurements. 

### 2.7. Statistical Analysis

Statistical analyzes were performed using IBM SPSS Statistics 26.0. In order to compare the two groups with each other, the Mann-Whintey U test was carried out. Wilcoxon’s test was performed to compare the results on different platforms/intra-group test. The level of significance was α = 0.05.

## 3. Results

### 3.1. Static Balance

In order to compare the studied groups, an analysis was carried out using the Mann-Whitney U test in terms of the parameters measured. The analysis showed that stroke patients scored higher for anterior, posteriori, medial and lateral for both eyes closed and eyes open. The results are presented in Table 3.

Analysis with the Wilcoxon test performed separately for each group showed that both among patients after stroke (*p* = 0.012) and healthy (*p* = 0.012), significantly higher results were recorded for the measurement with eyes closed compared to the measurement with eyes open for all the measured parameters: anterior, posteriori, medial and lateral.

### 3.2. Dynamic Balance

Similar calculations with the Mann Whitney U test were carried out in both study groups for the results of the study on an unstable platform (fall risk test). The analysis showed that patients after stroke obtained higher results for frontal tilt with eyes closed than healthy patients. For the remaining measurements, the differences between the groups turned out to be insignificant (Table 4).

Eyes open vs eyes closed—Detailed analysis of the results with the Wilcoxon test in each group showed that among stroke patients with eyes closed, significantly higher results were recorded only for the measurement of inclination in the frontal plane compared to the measurement with eyes open. Sagittal differences were not significant. On the other hand, healthy people in both planes obtained significantly higher results with eyes closed (Table 5).

### 3.3. Assessment of Muscle Tension

In order to compare the two groups, the Mann Whintey U test was performed in terms of the parameters measured. The analysis showed that stroke patients during the Unterberger test had higher SCM voltage scores with eyes closed than healthy patients. In the studies on the stable platform, significant differences occurred for the LC muscle tension both with eyes closed and open; patients after stroke obtained higher results compared to healthy patients. In studies on the unstable platform, stroke patients obtained higher results for SCM tension with eyes closed and open compared to healthy patients. The results are presented in Table 6.

#### 3.3.1. Eyes Open vs Eyes Closed

Detailed analysis of the results using the Wilcoxon test in each of the studied groups showed that among patients after a stroke with eyes closed. they obtained significantly higher results for SCM during the Unterberger test (*p* = 0.012), for the longus colli muscle in the studies on the stable platform (*p* = 0.017), and SCM and LC in the study on the unstable platform (*p* = 0.012). The strength of the effect was great. In the group of healthy people. significant differences occurred only in the Unterberger test (for SCM *p* = 0.025; LC *p* = 0.012)—higher results were obtained with eyes closed compared to the measurement with eyes open. For the measurements on the stable and unstable platforms. the differences between the measurements with eyes closed and open turned out to be insignificant.

#### 3.3.2. Unterberger Test vs. Stable Platform (PS)

Analogous analyzes were performed to compare the results during the Unterberger test and on the stable platform. The analysis showed no significant differences in the obtained results among people from the group after stroke. both for measurements with eyes closed and eyes open.

Among healthy people. the Unterberger test obtained significantly higher results for LC muscle tone. both with eyes open (*p* = 0.036) and eyes closed (*p* = 0.012). The strength of the effect was great.

#### 3.3.3. Unterberger Test vs. Unstable Platform (PNS)

Then the results obtained in the Unterberger test and on the unstable platform were compared with each other. The analysis showed that among the stroke patients on the unstable platform. the eyes-open measurement obtained higher results for SCM (*p* = 0.036) (strong effect). With eyes closed. no differences were noted.

Among healthy patients. the measurement in the Unterberger test for closed eyes obtained a significantly higher result for the LC muscle tension (*p* = 0.012). compared to the measurement on the unstable platform. The effect was strong.

#### 3.3.4. PS vs. PNS

Among post-stroke patients. higher SCM results were obtained for the measurement on the unstable platform. both in the test with eyes open (*p* = 0.025) and eyes closed (*p* = 0.017) (strong effects).

In the group of healthy patients. for the measurement on the unstable platform as compared to the stable platform. a higher result was obtained for LC with eyes closed (*p* = 0.043). a strong effect.

## 4. Discussion

The aim of the study was to assess the static and dynamic balance of the body and the tension of selected muscles of the cervical spine with and without visual control. Patients after stroke in the course of cerebral amyloid angiopathy (CAA) were studied in relation to healthy subjects. This assessment was made on the basis of the analysis of the signal from the balance platform and the tension of the sternocleidomastoid muscle and the longus colli muscle with superficial EMG during tests with open and closed eyes as well as the study of muscle tone and the importance of the organ of vision in the Unterberger qualitative test [39]. Our research has shown that patients with neurological disorders have higher amplitudes for the sternocleidomastoid muscle during dynamic tests (Unterberger and unstable platform). Healthy volunteers showed increased activity of both examined muscles only in the Unterberger test. As Fujiwara or Basmajian wrote. increased sternocleidomastoid activity may be related to the mechanics of this muscle. Increased rotation of the cervical spine and lateral displacement. due to the maintenance of postural balance. increases the activity of the sternocleidomastoid muscles [55]. According to Vitti et al. turning the head to the left and returning to the neutral position increases muscle activity in the right SCM. Conversely. right rotation and return to neutral increase activity in the left sternocleidomastoid muscle [55]. In contrast. during the stability test. the longus colli muscle was more active in the group of patients. both with eyes open and closed. In the control group. both examined muscles showed significantly higher activity only in the Unterberger test. Detailed group analysis showed that in the group of patients in the unstable platform test. the longus colli muscle was more active with eyes closed.

The greater activity of the longus colli muscle during the fall risk test is consistent with Mayoux-Benhamou et al. theory that the longus colli muscle plays one of the main roles in maintaining balance in the sagittal plane. When looking straight ahead. the weight of the head has a forward tilting effect and tends to increase cervical lordosis. The muscles of the back of the neck must then be more active in an upright position. increasing their activity also when the head is partially flexed to prevent the head tilting forward [56]. Additionally. the longus colli muscle (LC) is unique: it is the only muscle in the spine that is in front of the spine and has attachments limited to the vertebrae. Therefore. its action must be limited to the cervical spine—in contrast to the other anterior muscles of the head and neck. which act more or less indirectly [55,56,57]. The results obtained in our study indicate that static and dynamic imbalances concern especially people with greater functional deficits. but postural instability may also occur in static conditions in healthy people. The above disorders are associated with greater activity of the muscles of the cervical spine. According to our research. under the conditions of statics. the muscle that bends the cervical spine plays a dominant role and under the conditions of dynamics. the sternocleidomastoid muscle (with open eyes) plays the role of straightening the cervical spine. Turning off the organ of vision increases the activity of the muscles. flexors and extensors of the cervical spine. The activity of these muscles manifests itself more clearly. the greater the functional deficits we observed in patients. What’s more. they can also appear in healthy people.

In a posturographic study, Norre et al. showed that the cervical spine contains many mechanoreceptors responsible for maintaining balance. According to Osiński et al. cervical dizziness may result from damage to the intervertebral joints and paraspinal muscles. which in turn may lead to imbalances [58]. Cervical spine dysfunctions require a thorough imaging diagnosis [59]. Sipko et al. believed that states of cervical overload led to restricted mobility and the formation of an upper cross syndrome. According to them. this is of great importance in the event of imbalance in the eyes-closed test. This analysis indicates that the lack of visual control generally contributes to the imbalance [60,61].

As our results showed. eyesight plays an important role in maintaining the body’s balance. The correct interaction of the musculoskeletal system, eyesight and the balance system allow you to control body posture. Especially in the elderly. the interaction of these elements is important due to the risk of falls [62,63,64].

### 4.1. Research Value

The value of our study lies in the fact that so far. people with cerebral amyloid angiopathy have not been studied in terms of body balance. the importance of vision and changes in the tone of the muscles of the cervical spine and this may be of great importance in restoring fitness in people after a stroke in the course of CAA. In addition. we present the results of studies of people who have suffered a stroke of the cerebellum. less common in the course of CAA.

### 4.2. Study Limitation

However. work has limitations. First of all. the size of the groups was relatively small. The men from the stroke group are older than the healthy volunteers. which can be a problem with validating the results. However. the proportions of age. sex. height and weight as well as BMI in the studied groups were maintained and the lack of age uniformity in the male and female groups is mainly caused by the size of the sample.

Moreover. in our research we compared sick people to healthy people. In future studies. we plan to compare stroke patients with CAA and other causes of stroke.

## 5. Conclusions

Patients diagnosed with CAA present significantly greater imbalances in terms of both statics and dynamics comparing to healthy subjects.

In static tests of CAA patients, the longus cervicis muscle was more active and in dynamic tests. the sternocleidomastoids muscle.

Eyesight plays an important role in controlling the balance. especially in healthy.

The more functional deficits the more difficult it is to restore balance also with eye control.

## Figures and Tables

**Table 1 diagnostics-11-02036-t001:** The epidemiological data of post stroke (CAA) and healthy control groups.

Total Number of Patients *n* = 16 (100%)	Post-Stroke CAA *n* = 8 (50%)		Healthy Controls *n* = 8 (50%)	
sex	Female	Male	Female	Male
n/%	6 (75%)	2 (25%)	6 (75%)	2 (25%)
Age range, years	55–68	57–90	56–82	56–58
Cerebral ischemic stroke n/%	3 (37.5%)		N/A	
Cerebral hemorrhage stroke n/%	2 (25%)		N/A	
Cerebellum ischemic stroke n/%	3 (37.5%)		N/A	
Time post stroke (week); acute	7–9		N/A	
Right affected side	5 (62.5%)		N/A	
Left affected side	3 (37.5%)		N/A	

**Table 2 diagnostics-11-02036-t002:** Biometric data of CAA population and healthy control groups.

Group	Age	Height	Body Mass	BMI
Stroke CAA ± SD	65.38 ± 11.34	168.25 ± 5.87	72.00 ± 6.48	25.44 ± 1.88
Control ± SD	64.50 ± 9.83	167.88 ± 9.11	72.63 ± 10.21	25.78 ± 2.88
Wilcoxon U	30.50	29.00	28.00	32.00
Z	−0.16	−0.32	−0.42	0.00
*p*	0.874	0.752	0.674	1.000
effect size	0.04	0.08	0.11	<0.01

**Table 3 diagnostics-11-02036-t003:** Comparison of stroke patients and healthy patients in terms of the parameters measured.

	Stroke (*n* = 8)				Healthy (*n* = 8)				Z	*p*	r
	Average Rank	M	Me	SD	Average Rank	M	Me	SD			
**Eyes open**											
Anterior	11.50	1.01	1.09	0.37	5.50	0.43	0.38	0.13	−2.52	0.012	0.63
Posterior	11.50	0.86	0.86	0.34	5.50	0.43	0.42	0.08	−2.52	0.012	0.63
Medial	11.63	0.53	0.60	0.19	5.38	0.20	0.20	0.07	−2.63	0.009	0.66
Lateral	11.38	0.49	0.55	0.18	5.63	0.21	0.21	0.07	−2.42	0.016	0.60
**Eyes closed**											
Anterior	12.50	1.38	1.44	0.17	4.50	0.65	0.58	0.17	−3.36	0.001	0.84
Posterior	12.50	1.26	1.25	0.30	4.50	0,59	0.59	0.06	−3.36	0.001	0.84
Medial	11.13	0.97	1.04	0.39	5.88	0.46	0.47	0.08	−2.21	0.027	0.55
Lateral	11.25	0.91	0.93	0.43	5.75	0.37	0.39	0.09	−2.31	0.021	0.58

**Table 4 diagnostics-11-02036-t004:** Comparison of stroke patients and healthy patients in terms of the parameters measured.

	Stroke (*n* = 8)				Healthy (*n* = 8)				Z	*p*	r
	Average Rank	M	Me	SD	Average Rank	M	Me	SD			
**Eyes open**											
unstable platform 2/6 frontal	10.19	4.69	4.60	1.46	6.81	3.79	3.50	1.64	−1.42	0.156	0.35
unstable platform 2/6 sagittal	7.88	3.46	2.56	1.41	9.13	3.46	3.03	1.28	−0.53	0.600	0.13
**Eyes closed**											
unstable platform 2/6 frontal	11.31	6.38	6.20	1.09	5.69	4.70	4.55	1.61	−2.36	0.018	0.59
unstable platform 2/6 sagittal	8.50	4.16	4.07	0.91	8.50	4.28	3.84	1.30	0.00	1.000	0.00

**Table 5 diagnostics-11-02036-t005:** Comparison of the results of measurements with eyes closed and eyes open in the group of patients after stroke.

	Eyes Open			Eyes Closed			Z	*p*	r
	M	Me	SD	M	Me	SD			
**Stroke**									
unstable platform 2/6 frontal	4.69	4.60	1.46	6.38	6.20	1.09	−2.52	0.012	0.63
unstable platform 2/6 sagittal	3.46	2.56	1.41	4.16	4.07	0.91	−1.12	0.263	0.63
**Healthy**									
unstable platform 2/6 frontal	3.79	3.50	1.64	4.70	4.55	1.61	−2.52	0.012	0.63
unstable platform 2/6 sagittal	3.46	3.03	1.28	4.28	3.84	1.30	−2.52	0.012	0.63

**Table 6 diagnostics-11-02036-t006:** Comparison of stroke patients and healthy patients in terms of the parameters measured.

	Stroke (*n* = 8)				Healthy (*n* = 8)				Z	*p*	r
	Average Rank	M	Me	SD	Average Rank	M	Me	SD			
**Unterberger test**											
(EO) tension SCM	10.63	13.13	13.24	8.79	6.38	7.45	7.88	3.65	−1.79	0.074	0.45
(EO) tension LC	8.00	19.23	16.96	13.53	9.00	22.28	17.85	11.88	−0.42	0.674	0.11
(EC) tension SCM	10.88	20.23	18.38	11.45	6.13	10.34	9.85	3.34	−2.00	0.046	0.50
(EC) tension LC	7.88	23.70	22.12	12.12	9.13	27.16	23.67	14.07	−0.53	0.600	0.13
**Stable platform (PS)**											
(EO) tension SCM	10.25	20.43	13.32	16.99	6.75	10.43	9.21	5.50	−1.47	0.141	0.37
(EO) tension LC	11.38	23.85	19.09	11.29	5.63	16.14	14.35	5.12	−2.42	0.016	0.60
(EC) tension SCM	10.50	20.17	15.30	13.18	6.50	10.63	10.02	5.18	−1.68	0.093	0.42
(EC) tension LC	12.13	25.14	21.21	11.08	4.88	14.98	14.54	2.68	−3.05	0.002	0.76
**Unstable platform (PNS)**											
(EO) tension SCM	11.13	22.81	15.86	16.23	5.88	10.27	8.36	5.29	−2.21	0.027	0.78
(EO) tension LC	10.38	24.93	22.40	10.25	6.63	17.75	17.55	6.60	−1.58	0.115	0.56
(EC) tension SCM	11.38	24.10	18.07	16.10	5.63	10.40	8.85	5.20	−2.42	0.016	0.85
(EC) tension LC	10.75	26.47	24.38	10.36	6.25	18.03	17.83	5.62	−1.89	0.059	0.47

## Data Availability

Data available on request from corresponding author.

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
