# Peer review of "The Relationship between the Static and Dynamic Balance of the Body, the Influence of Eyesight and Muscle Tension in the Cervical Spine in CAA Patients—A Pilot Study"

_diagnostics, 2021, doi:10.3390/diagnostics11112036_

Round 1
Reviewer 1 Report
I read with interest this study regarding the relationship between the static and dynamic balance of the body, the influence of eyesight and muscle tension in the cervical spine in CAA patients.
The study is overall well written. The casuistry is small, and this should be considered when drawing conclusions and stated in limitations.
There are some points that need to be corrected:
- I suggest anyway a revision by a native English speaking person with medical background.
- The abstract is not clear in the second half.
- you could add a reference in discussion regarding imaging of the muscle in the spine (for example: Colosimo C et al. Imaging in degenerative spone pathology. )
Author Response
Manuscript ID: diagnostics-1422647
Type of manuscript: Article
Title: The relationship between the static and dynamic balance of the body, the influence of eyesight and muscle tension in the cervical spine in CAA patients - a pilot study.
Dear Reviewers,
Thank you very much for analysing our manuscript.
We greatly appreciate your comments and the indication of passages that should be corrected and clarified. Taking into account your suggestions, all mistakes have been corrected. The introduction of corrections and changes to the text may have resulted in a shift in the numbering of lines. To avoid confusion, changes made to the text are marked in blue, and the manuscript has been uploaded in the track changes mode.
Reviewer #1:
Thank you very much for the very quick and thorough analysis of our manuscript.
Thank you for such a favourable analysis of our manuscript. It is a great joy for us to hear that our manuscript is assessed as very interesting.
The following comments and answers:
I suggest anyway a revision by a native English speaking person with medical background.
The work has been reviewed and revised by a native medical English translator.
I hope that the text, as it stands today, meets the requirements.
Thank you for pointing this out.
The abstract is not clear in the second half.
The summary of the work has also been improved. The correction applies in particular to the purpose of the research and the methodology. The current form of the summary seems to represent the essence of the work.
Thank you very much for this comment.
You could add a reference in discussion regarding imaging of the muscle in the spine (for example: Colosimo C et al. Imaging in degenerative spone pathology. Acta Neurochirurgica, Supplementum 2011, (108), pp. 9–15)
In response to your review, I enriched the discussion a bit about the possibility of diagnosing the cervical spine. As a result, another item appeared in the list of references.
Thank you so much for your review and suggestion.
Thank you very much for your time.
Reviewer 2 Report
I ask you to value the following suggestions for improving the paper
Introduction:
The epidemiological data are from 1998, it would be necessary to present updated and contextualized data within the world population.
The subjets begins to be described in the introduction, where it is noted that the cerebellar lesion is rare and that, nevertheless, it is present in the work (3 subjects out of 8, from the stroke group). It is recommended that this information does not appear in the introduction and, since the sample was not randomly selected, it should have been included in the inclusion and exclusion criteria.
The stated decision on the decision to suppress the vision must appear in the methodology and justified in the introduction.
The objectives of the research are not sufficiently defined, it would be convenient to expose the balance aspects that are going to be studied.
Methology
It does not appear when the measurement was made or what was the process to select the subjects.
Men in the stroke group are older than those in the control group. This can be a problem to validate the results
Author Response
Manuscript ID: diagnostics-1422647
Type of manuscript: Article
Title: The relationship between the static and dynamic balance of the body, the influence of eyesight and muscle tension in the cervical spine in CAA patients - a pilot study.
Dear Reviewers,
Thank you very much for analysing our manuscript.
We greatly appreciate your comments and the indication of passages that should be corrected and clarified. Taking into account your suggestions, all mistakes have been corrected. The introduction of corrections and changes to the text may have resulted in a shift in the numbering of lines. To avoid confusion, changes made to the text are marked in blue, and the manuscript has been uploaded in the track changes mode.
Reviewer #2:
Thank you very much for the very quick and thorough analysis of our manuscript.
The following comments and answers:
Introduction:
The epidemiological data are from 1998, it would be necessary to present updated and contextualized data within the world population.
In response to your review, I have revised the bibliography with regard to CAA epidemiology. To illustrate the CAA prevalence statistics, I used the publication from 2011, in which the author presents the disease incidence depending on the age of the respondents. Therefore, a new item has appeared in the list of references:
- Arvanitakis Z, Leurgans SE, Wang Z, et al. Cerebral amyloid angiopathy pathology and cognitive domains in older persons. Ann Neurol 2011; 69:320.
Thank you very much for drawing your attention to this issue.
The subjects begins to be described in the introduction, where it is noted that the cerebellar lesion is rare and that, nevertheless, it is present in the work (3 subjects out of 8, from the stroke group). It is recommended that this information does not appear in the introduction and, since the sample was not randomly selected, it should have been included in the inclusion and exclusion criteria.
Indeed, the text on the analysis of cerebellar stroke patients was not needed in the introduction. This text has been deleted and the sentence revised.
Indeed, the text on the analysis of cerebellar stroke patients was not needed in the introduction. This text has been deleted and the sentence revised.
Selection to the test group and to the control group was targeted and this information was supplemented with the criteria for including patients and healthy people.
Thank you very much for this review and suggestion.
The stated decision on the decision to suppress the vision must appear in the methodology and justified in the introduction.
Both in the introduction and in the methodology of the research and in the discussion, the importance of eye control for the possibility of maintaining static and dynamic balance and for changes in the tension of the muscles of the cervical spine was emphasized. Especially in the introduction and discussion, the authors argue that turning off the eyesight shows the effectiveness of the other balance control mechanisms. All changes are highlighted in blue and presented in the track changes mode.
Thank you for this comment.
The objectives of the research are not sufficiently defined, it would be convenient to expose the balance aspects that are going to be studied.
The purpose of the work was precisely defined. I emphasized what was examined (static and dynamic balance of the body and muscle tension in the cervical spine), depending on the control or lack of control of the organ of vision.
Thank you very much for your attention and suggestion.
Methodology
It does not appear when the measurement was made or what was the process to select the subjects.
The research methodology was supplemented with information on and the period of time when the research was performed and the process of selecting the respondents.
Thank you very much for this comment.
Men in the stroke group are older than those in the control group. This can be a problem to validate the results
Indeed, the men in the stroke group are older than the healthy volunteers, and this can be a problem with validating the results. However, I would like to point out that the proportions of age, sex, height and weight are maintained in the studied groups. The lack of age uniformity in the male and female gender groups is mainly due to the size of the sample.
Taking into account your comment, I increased the number of study limitation.
Thank you very much for this review.
Thank you very much for your time.
Round 2
Reviewer 2 Report
The proposed changes have been incorporated, improving the quality of the manuscript. It is recommended to review the bibliography so that it has a homogeneous format in all the references